Genome-wide analysis of 3′-untranslated regions supports the existence of post-transcriptional regulons controlling gene expression in trypanosomes

De Gaudenzi Javier G. jdegaudenzi@iib.unsam.edu.ar
Carmona Santiago J.
Agüero Fernán
Frasch Alberto C.
Instituto de Investigaciones Biotecnológicas-Instituto Tecnológico de Chascomús, UNSAM-CONICET , Buenos Aires , Argentina
Salsbury Jr Freddie
Electronic publication date: 2013 Jul 30
Publication date: 2013
Volume: 1
Electronic Location ID: e118
Received 2013 Jun 5; Accepted 2013 Jul 10
Copyright: © 2013 De Gaudenzi et al.
Copyright year: 2013
Copyright holder: De Gaudenzi et al.
License: This is an open access article distributed under the terms of the Creative Commons Attribution License, which permits unrestricted use, distribution, and reproduction in any medium, provided the original author and source are credited.
License URL: https://creativecommons.org/licenses/by/3.0/

Keywords: RNA-binding protein, Trypanosomes, Cis-element, Post-transcriptional control, RNA regulon

Funding: Agencia Nacional de Promoción Científica y Tecnológica (ANPCyT) Consejo Nacional de Investigaciones Científicas y Técnicas (CONICET) Fundación Bunge y Born The work described in this article was performed with financial support from the Agencia Nacional de Promoción Científica y Tecnológica (ANPCyT) to JGDG and ACF, the Consejo Nacional de Investigaciones Científicas y Técnicas (CONICET) to JGDG, and the Fundación Bunge y Born to JGDG. JGDG, FA and ACF are members of the Research Career of CONICET, and SC is a CONICET Research Fellow. The funders had no role in study design, data collection and analysis, decision to publish, or preparation of the manuscript.

==============================
In eukaryotic cells, a group of messenger ribonucleic acids (mRNAs) encoding functionally interrelated proteins together with the trans-acting factors that coordinately modulate their expression is termed a post-transcriptional regulon, due to their partial analogy to a prokaryotic polycistron. This mRNA clustering is organized by sequence-specific RNA-binding proteins (RBPs) that bind cis-regulatory elements in the noncoding regions of genes, and mediates the synchronized control of their fate. These recognition motifs are often characterized by conserved sequences and/or RNA structures, and it is likely that various classes of cis-elements remain undiscovered. Current evidence suggests that RNA regulons govern gene expression in trypanosomes, unicellular parasites which mainly use post-transcriptional mechanisms to control protein synthesis. In this study, we used motif discovery tools to test whether groups of functionally related trypanosomatid genes contain a common cis-regulatory element. We obtained conserved structured RNA motifs statistically enriched in the noncoding region of 38 out of 53 groups of metabolically related transcripts in comparison with a random control. These motifs have a hairpin loop structure, a preferred sense orientation and are located in close proximity to the open reading frames. We found that 15 out of these 38 groups represent unique motifs in which most 3′-UTR signature elements were group-specific. Two extensively studied Trypanosoma cruzi RBPs, TcUBP1 and TcRBP3 were found associated with a few candidate RNA regulons. Interestingly, 13 motifs showed a strong correlation with clusters of developmentally co-expressed genes and six RNA elements were enriched in gene clusters affected after hyperosmotic stress. Here we report a systematic genome-wide in silico screen to search for novel RNA-binding sites in transcripts, and describe an organized network of several coordinately regulated cohorts of mRNAs in T. cruzi. Moreover, we found that structured RNA elements are also conserved in other human pathogens. These results support a model of regulation of gene expression by multiple post-transcriptional regulons in trypanosomes.

Introduction

The kinetoplastid protozoa comprise a group of unicellular parasites that belong to a distinctive evolutionary lineage of eukaryotes. Members of this taxonomic group include etiological agents of several neglected zoonoses such as Chagas disease (Trypanosoma cruzi), sleeping sickness (Trypanosoma brucei) and Leishmaniasis (Leishmania spp.). These three species are digenetic unicellular microorganisms that suffer continuous morphological changes throughout their complex life-cycles (Barrett et al., 2003).

Transcription in these cells is polycistronic. RNA synthesis by RNA polymerase II starts at a few genomic locations within chromosomes and thus nearly all the protein-coding genes are arrayed in long multi-gene transcription units (Fernandez-Moya & Estevez, 2010; Kramer, 2011). In contrast to operons in bacteria, trypanosomal polycistronic units require processing before translation. Consequently individual mature messenger ribonucleic acids (mRNAs) are generated by 5′ trans-splicing and 3′ polyadenylation of precursor RNAs (Hendriks & Matthews, 2007). Given these unique genetic features, trypanosomes essentially make use of post-transcriptional processes to control gene expression [reviewed in (De Gaudenzi et al., 2011)].

A common polypyrimidine tract located between two neighboring open reading frames is the signal sequence recognized by both trans-splicing and polyadenylation machineries and governs the co-transcriptional RNA processing (Matthews, Tschudi & Ullu, 1994). A 39-nt capped spliced-leader sequence is added only a few nucleotides upstream of the ATG translational start codon thus generating short 5′-UTRs. Because this region is usually under rigid structural constraints to accommodate the translational machinery (Conne, Stutz & Vassalli, 2000), the 3′-UTR is usually the key region involved in transcript stability and translation efficiency.

Bioinformatic tools allowed the identification of all trypanosomal RBPs and numerous sequence elements mainly involved in RNA-processing and genome structure (Benz et al., 2005; Campos et al., 2008; Duhagon, Dallagiovanna & Garat, 2001; Duhagon et al., 2013; Smith, Blanchette & Papadopoulou, 2008). Several studies demonstrated the presence of U-rich elements in trypanosomal mRNA 3′-UTRs [reviewed in Araujo & Teixeira, 2011; Haile & Papadopoulou, 2007; Hendriks & Matthews, 2007]. Strikingly, the functional role of CA repeated tracts in T. cruzi 3′-UTRs was recently established as a signal for gene expression modulation through the parasite’s life-cycle (Pastro et al., 2013).

Cis-acting motifs are recognized by different trans-acting factors, including members of the kinetoplastid superfamily of RNA-recognition motif (RRM)-containing RNA-binding proteins (RBPs) (Kramer & Carrington, 2011). The first two RRM proteins of this family were previously characterized by our group and termed T. cruzi U-rich RBP 1 (TcUBP1) and TcUBP2. A third member of this group was named TcRBP3 and displayed different RNA-binding properties than the previously mentioned RBPs (De Gaudenzi, D’Orso & Frasch, 2003). A comparative ribonomic analysis of TcUBP1 and TcRBP3 showed that both proteins can share target transcripts, although they preferentially bind different sets of mRNAs. These trypanosomal target transcripts were classified within functional groups and contain conserved structural elements involved in RNA-binding in their 3′-UTRs (Noe, De Gaudenzi & Frasch, 2008). Furthermore, these two RRM-containing RBPs can associate with more than one RNA element within the same transcript, supporting the idea that the combination of motifs is the main factor that defines RNA-protein interaction networks (Mittal et al., 2009; Morris, Mukherjee & Keene, 2010).

A group of functionally linked mRNAs together with the sequence-specific RBPs that coordinately modulate their expression is termed an RNA regulon, due to their partial analogy to the bacterial operon. The messenger ribonucleoprotein (mRNP) complex-driven organization of transcripts allows eukaryotic cells to control protein synthesis from genes that are dispersed throughout the genome but encode for products involved in common or related functions. This higher-order cytoplasmic organization of transcripts implies a complex but flexible level of gene regulation that entails a rapid adaptation of the cellular transcriptome in response to alterations in the environment (Mansfield & Keene, 2009). Post-transcriptional regulons have been described in mammalian cells, fruit flies, and yeast, and can control several associated processes such as RNA processing, export, stabilization, localization and translation. Moreover, regulons are important in different cellular pathways such as oxidative metabolism, stress response and circadian rhythms (Keene, 2007).

We and other authors demonstrated that Keene’s model of RNA regulons precisely fits the observed trypanosome gene expression regulation (Noe, De Gaudenzi & Frasch, 2008; Ouellette & Papadopoulou, 2009; Queiroz et al., 2009). Genomes analysis of TriTryps provided a large collection of putative RBPs and mRNA metabolism factors, but an extensive characterization of RNA-protein interactions still remains elusive (Kramer, Kimblin & Carrington, 2010). This is due, at least in part, because the cis-elements that orchestrate these interactions are poorly defined. In particular, more efforts are necessary to complete the global identification of conserved sequences that govern large cohorts of trypanosome stage-regulated mRNAs. Post-transcriptional gene regulation does not seem to be exclusively governed by linear motifs, thus models of RNA-protein interactions should include both primary sequence and secondary structures features (Goodarzi et al., 2012).

An exhaustive genome-wide computational search for regulatory RNA elements has been reported in T. brucei (Mao, Najafabadi & Salavati, 2009) and conserved intercoding sequences and putative regulons were also identified in Leishmania (Vasconcelos et al., 2012). The observation that RRM-type RBPs recognize conserved structural motifs located in the 3′-UTR from functionally related targets, prompted us to search the T. cruzi genome in order to systematically describe the elements defining RNA regulons. We found that distinct groups of metabolically clustered transcripts contain cis-regulatory signals. These cis-elements have stem-loop secondary structures, and were preferentially located in the 3′-UTR of transcripts (but not in the 5′-UTR), with a particular sense orientation at the vicinity of the coding sequence. Here we describe, for the first time, a systematic identification of candidate RNA regulons in kinetoplastids grouped by similar metabolic pathways, and harboring signature structured RNA motifs. The identification of shared elements in cohorts of transcripts will pave the way for the detection of the trans-acting factors that organize each group of mRNAs and govern their final behaviour. All these observations are consistent with the RNA regulon model.

Results

Conserved structural RNA elements in 3′-UTRs of mRNAs encoding metabolically interrelated proteins of T. cruzi

Using a bioinformatic approach, we focused our work on the detection of potential structural cis-elements located in noncoding regions of functionally related transcripts. Towards this goal, we carried out a global analysis using T. cruzi genomic data (El-Sayed et al., 2005). Since there is no information available for T. cruzi RNA-seq reads, noncoding sequences have been inferred from average lengths of 5′- and 3′-UTRs of T. cruzi published transcripts (Brandao & Jiang, 2009; Campos et al., 2008) and extracted from TriTrypDB (http://tritrypdb.org/tritrypdb/) (Methods). These sequences were classified into functional categories in accordance to the KEGG pathway database (Kanehisa & Goto, 2000) (http://www.genome.jp/kegg/pathway.html). We next generated lists of putative 3′ noncoding regions for each KEGG class containing genomic sequences resembling 3′-UTRs. Allelic copies identified in the hybrid TcVI CL Brener genome having similar 3′-UTRs, 80% identity or higher, were filtered to reduce redundancy (see File S1 and Methods for details). As a result, we categorized the T. cruzi proteins within 80 groups including 1814 genes, but only those classes having at least 10 sequences were used in this paper. Thus, we limited our search to 53 categories termed tcr00010 to tcr04650 (see Table 1 for descriptions) which enclose 1617 total genes.

Table 1 Metabolic gene clusters used for motif elucidation.

List of the 53 T. cruzi metabolic groups obtained from KEGG pathway database composed by at least 10 sequences that were used for motif search. Overall, 43 out of 53 categories have at least 50% of the BLAST sequences containing the complete motif within the EST hit, reinforcing the idea that our datasets could be used to identify putative regulatory RNA elements.

KEGG group	Description	N	%	
tcr00010	Glycolysis/Gluconeogenesis	61	36.4	
tcr00020	Citrate cycle (TCA cycle)	37	44.4	
tcr00030	Pentose phosphate pathway	33	75.0	
tcr00051	Fructose and mannose metabolism	20	60.0	
tcr00071	Fatty acid metabolism	32	80.0	
tcr00100	Steroid biosynthesis	13	75.0	
tcr00130	Ubiquinone and other terpenoid-quinone biosynthesis	11	100.0	
tcr00190	Oxidative phosphorylation	63	80.0	
tcr00230	Purine metabolism	88	66.7	
tcr00240	Pyrimidine metabolism	69	60.0	
tcr00250	Alanine, aspartate and glutamate metabolism	26	88.9	
tcr00260	Glycine, serine and threonine metabolism	22	100.0	
tcr00270	Cysteine and methionine metabolism	26	50.0	
tcr00280	Valine, leucine and isoleucine degradation	37	33.3	
tcr00310	Lysine degradation	26	80.0	
tcr00330	Arginine and proline metabolism	26	25.0	
tcr00350	Tyrosine metabolism	13	100.0	
tcr00380	Tryptophan metabolism	24	0.0	
tcr00410	beta-Alanine metabolism	16	75.0	
tcr00450	Selenocompound metabolism	13	66.7	
tcr00480	Glutathione metabolism	33	70.0	
tcr00500	Starch and sucrose metabolism	12	85.7	
tcr00510	N-Glycan biosynthesis	17	0.0	
tcr00520	Amino sugar and nucleotide sugar metabolism	39	75.0	
tcr00561	Glycerolipid metabolism	12	71.4	
tcr00562	Inositol phosphate metabolism	19	100.0	
tcr00563	Glycosylphosphatidylinositol (GPI)-anchor biosynthesis	10	100.0	
tcr00564	Glycerophospholipid metabolism	15	100.0	
tcr00620	Pyruvate metabolism	39	63.6	
tcr00630	Glyoxylate and dicarboxylate metabolism	16	100.0	
tcr00640	Propanoate metabolism	20	100.0	
tcr00650	Butanoate metabolism	23	100.0	
tcr00670	One carbon pool by folate	10	100.0	
tcr00785	Lipoic acid metabolism	10	100.0	
tcr00900	Terpenoid backbone biosynthesis	12	80.0	
tcr00910	Nitrogen metabolism	14	0.0	
tcr00970	Aminoacyl-tRNA biosynthesis	36	50.0	
tcr03010	Ribosome	201	61.2	
tcr03018	RNA degradation	31	100.0	
tcr03020	RNA polymerase	23	83.3	
tcr03030	DNA replication	43	100.0	
tcr03040	Spliceosome	44	37.5	
tcr03050	Proteasome	50	0.0	
tcr03410	Base excision repair	31	100.0	
tcr03420	Nucleotide excision repair	41	100.0	
tcr03430	Mismatch repair	27	80.0	
tcr03440	Homologous recombination	19	100.0	
tcr03450	Non-homologous end-joining	12	100.0	
tcr04070	Phosphatidylinositol signaling system	26	14.3	
tcr04120	Ubiquitin mediated proteolysis	35	100.0	
tcr04130	SNARE interactions in vesicular transport	46	100.0	
tcr04144	Endocytosis	37	50.0	
tcr04650	Natural killer cell mediated cytotoxicity	12	100.0	
Notes.

N, number of sequences in each group. %, percentage of sequences harboring RNA motifs within the annotated EST hit in each metabolic group.

Linear motifs are reportedly difficult to define, especially in repeat-rich and atypical TriTryp genomes which contain pyrimidine-rich elements in the intergenic regions (El-Sayed et al., 2005; Hendriks & Matthews, 2007). Thus, we used the CMfinder software (Yao, Weinberg & Ruzzo, 2006) (http://bio.cs.washington.edu/yzizhen/CMfinder/) for structural RNA motif prediction in the putative 3′-UTR sequences of each group. Covariance models are RNA motif models that represent both the sequence and structure binding preferences of RBPs. We chose the best top ranked motif provided by the program. Therefore, 53 new RNA structural elements were identified and termed according to the number of the KEGG pathway from which the motifs were obtained: e.g., m00010 is the motif derived from the tcr00010 dataset (Glycolysis/Gluconeogenesis), m00020 from tcr00020 (Citrate cycle), etc. Figure 1 illustrates the motif discovery pipeline used (Fig. 1A) and a pie chart distribution of the metabolic groups having at least 10 genes used as the input data (Fig. 1B).

Figure 1 Computational workflow and dataset.

(A) Motif discovery pipeline used in this work. (B) Pie chart distribution of the 53 KEGG metabolic groups having at least 10 3′-UTR sequences used as input for RNA motif elucidation.

Figure 2 shows the RNA structures for the predicted motifs. Structured elements had a length ranging from 28 nts (tcr00240, Pyrimidine metabolism) to 87 nts (tcr03010, Ribosome). Nearly all the consensus motifs fold as a predicted stem-loop structure, with an average hairpin length of 15 bp and a loop ranging from 3 to 18 nts, giving rise to loops of a median length of 4 nts. Based on the logo representation, some motifs were classified according to their nucleotide composition. File S2 shows the consensus sequence, secondary structure in bracket notation and sequence logo of all the candidate RNA elements.

Figure 2 Conserved structural elements in predicted T. cruzi 3′-UTRs.

Secondary structures of the 53 conserved RNA elements were predicted using RNAfold program (Hofacker, 2003). Sequence composition of hairpin-loops having a singular nucleotide enrichment are shown below the panels. Sixteen loops of T. cruzi 3′-UTR motifs have an exclusive AU composition, six encompass the pattern UAUA, seven AUAU, and the others contain the AAU, AUUU or UUUUAU topology.

Evaluating the significance of the motif enrichment by randomization tests

Next, we further analyzed the specific-enrichment of the RNA elements in the KEGG groups. Consequently the motif representation was calculated as the percentage of element-containing sequences over the total number of sequences in each category (detailed under Methods). Overall, 79% of the groups have specific RNA elements. Accordingly, 42 out of 53 KEGG categories encompassed conserved structural motifs statistically enriched in their 3′-UTRs in comparison with control groups using random 3′-UTR datasets (Z-test, FDR 10%) (Fig. 3 and Table S1). For example, the RNA motif m00030 that was originally discovered in the 3′-UTRs of genes from the Pentose phosphate pathway (tcr00030) was detected in 69.7% of the sequences (23 of 33 mRNAs) but only in 26% ± 6% against 50 random searches.

Figure 3 Motif density in predicted T. cruzi 3′-UTRs.

Conserved structural motifs statistically enriched in several KEGG categories in comparison with 50 control random groups. The chart shows the percentage of motif-containing sequence in each group. Elements marked with a black circle have no statistical significance after considering the coverage test (χ2 test, FDR 5%) (see text).

These results reveal that ≈ 80% of the metabolic pathways analyzed contain specific signature elements in their noncoding regions different to what could be expected by chance in a group of random sequences. To see if these RNA motifs are specifically located at 3′-UTRs, we repeated our searches using the elements identified in the 3′-UTRs as queries against a 5′-end dataset. This dataset is composed of 350-nt fragments of the 5′-end of T. cruzi transcripts, grouped according to KEGG pathways (Methods). We next calculate the relative frequencies of elements (number of hits found in each group divided by its sequence length). Virtually all the motifs were noticeably over-represented in the 3′-UTRs compared with the 5′-end subset (Fig. S1), indicating a preferred 3′-UTR localization. See File S3 for a complete list of element-containing genes.

Constraints in motif orientation and position

RNA-binding elements need to be oriented in a particular direction relative to the coding sequence in order to carry out their regulatory function (Elemento, Slonim & Tavazoie, 2007). Thus, motif sequences are biologically significant when located on one strand, but not on the other. To evaluate sequence orientation preference, we compared the motif occurrences on the transcribed strand (genuine transcript) against the information acquired by its occurrences on the non transcribed strand (anti-sense transcript). The coverage test showed that most of the motifs previously described (38 out of 42) have an orientation bias, with a particular sense orientation (χ2 test, FDR 5%) (Table S2 and Fig. S2). This property is consistent with RNA regulatory motifs located in a precise orientation with respect to the coding sequence. A definitive list containing the motif’s representation of the 38 significant RNA elements and their P-values are shown in Table 2.

Table 2 List of structural RNA elements found in this work.

List of the 38 elements statistically enriched in the 3′-UTR of metabolically related transcripts, KEGG groups, motif coverages and P-values with FDR 5%.

RNA element	KEGG group	Cov. (%)	P-value	
m00020	Citrate cycle (TCA cycle)	64.9	1.4E−03	
m00030	Pentose phosphate pathway	69.7	8.3E−06	
m00071	Fatty acid metabolism	84.4	4.7E−05	
m00240	Pyrimidine metabolism	39.1	3.1E−02	
m00250	Alanine, aspartate and glutamate metabolism	65.4	1.4E−03	
m00260	Glycine, serine and threonine metabolism	63.6	2.3E−08	
m00270	Cysteine and methionine metabolism	65.4	7.7E−02	
m00280	Valine, leucine and isoleucine degradation	78.4	2.1E−03	
m00310	Lysine degradation	80.8	7.7E−10	
m00330	Arginine and proline metabolism	65.4	3.4E−04	
m00380	Tryptophan metabolism	75.0	2.9E−03	
m00410	beta-Alanine metabolism	68.8	1.2E−02	
m00480	Glutathione metabolism	33.3	0.0E + 00	
m00500	Starch and sucrose metabolism	91.7	3.3E−02	
m00510	N-Glycan biosynthesis	82.4	1.3E−08	
m00520	Amino sugar and nucleotide sugar metabolism	76.9	1.4E−12	
m00561	Glycerolipid metabolism	100.0	5.1E−06	
m00562	Inositol phosphate metabolism	73.7	7.8E−10	
m00564	Glycerophospholipid metabolism	73.3	2.0E−05	
m00620	Pyruvate metabolism	87.2	2.3E−08	
m00630	Glyoxylate and dicarboxylate metabolism	87.5	7.4E−05	
m00640	Propanoate metabolism	90.0	9.7E−04	
m00670	One carbon pool by folate	80.0	4.9E−05	
m00785	Lipoic acid metabolism	100.0	0.0E + 00	
m00900	Terpenoid backbone biosynthesis	83.3	1.3E−10	
m00910	Nitrogen metabolism	42.9	5.3E−06	
m00970	Aminoacyl-tRNA biosynthesis	66.7	2.0E−06	
m03010	Ribosome	92.5	0.0E + 00	
m03020	RNA polymerase	78.3	1.0E−01	
m03030	DNA replication	48.8	6.4E−03	
m03050	Proteasome	52.0	4.5E−03	
m03420	Nucleotide excision repair	51.2	3.5E−02	
m03430	Mismatch repair	66.7	3.1E−03	
m03440	Homologous recombination	78.9	0.0E + 00	
m04070	Phosphatidylinositol signaling system	46.2	2.3E−10	
m04120	Ubiquitin mediated proteolysis	51.4	4.0E−08	
m04130	SNARE interactions in vesicular transport	80.4	1.5E−12	
m04144	Endocytosis	64.9	4.1E−08	

Because there is no data available for the full T. cruzi transcriptome, we searched a trypanosomal transcript database for the presence of Expressed Sequence Tag records that could match the predicted 3′-UTR used in this work. About 10% of the total motif-containing KEGG sequences (107 out of 1090 non-redundant genes) gave a positive blast hit in the database considering an alignment of more than 150 nts with >95% of identity. Remarkably, 78.5% (84 out of 107) of these sequences contain the candidate element fully aligned against the EST transcript sequence. Table S3 shows a chart with the BLAST output of the KEGG genes indicating the motif position in the predicted 3′-UTR. Next, we sought to determine the percentage of sequence-containing motifs within the EST for each of the 53 KEGG metabolic group. As shown in Table 1 and Fig. S3, most of the KEGG groups analyzed here (43 out of 53) contain more than 50% of the sequences having the entire RNA candidate element aligned within the ESTdb hit; thus reinforcing the idea that the motifs are located within UTRs.

In this regard, we found that the majority of the motifs that were identified in 3′-UTRs have a position bias. If we split the 350-nt sequence into two segments, almost 60% of the elements were preferentially located proximal to the CDS. The proportion of RNA motifs located in the vicinity of the ORF increase to 75% when we restricted the study to those elements having the most significant e-values (E < 10−9) (Table S4). This observation has also been reported for some regulatory elements (Elemento, Slonim & Tavazoie, 2007; Mao, Najafabadi & Salavati, 2009). In Fig. 4A we show the distribution of RNA element localizations within the 3′-UTRs. However, this distribution is dependent on how we bin data. Contrarily, the probability density plot in Fig. 4B, which produces a smoother representation of the histogram, is independent of bin size. Both Figs. 4A and 4B show a declining trend for the localization of the identified RNA motifs as we move away from the stop codon. We are aware that this declining trend (observed at both ends in Fig. 4B) may reflect an artifactual boundary effect. However, because this effect is noticeably more pronounced in the distal part of the molecule (300–350 nts) we interpret this to suggest that the optimal localization of these motifs could be proximal to the coding region.

Figure 4 RNA motifs are located in the vicinity of the translational stop codon.

(A) Histogram of motif localization within the 3′-UTR. The distribution of the RNA elements (center position) in the 1335 genes from the KEGG categories were plotted with a bin width of 20 nts. (B) Probability density function of the 3′-UTR motif localization (Kernel density estimation, using Gaussian approximation for bandwidth selection).

Altogether, these results suggest that most of the sequences selected for this study could be considered as 3′ noncoding regions, with the RNA motifs most probably making part of the 3′-UTR and located at a particular distance from the stop codon, increasing the probability that the reported motifs have biological functions.

Motif representation in other metabolic pathways indicates that most of the RNA elements belong to a specific group

We next performed an all-against-all comparison of the presence of each motif in the 3′-UTRs of all KEGG categories, to evaluate the distribution of the previously identified elements in different metabolic groups. In this strategy, each RNA element is used to search into all individual KEGG T. cruzi datasets (e.g. the m00010 element is separately searched against dataset tcr00010 to tcr04650) (see Table S5). However this examination is not valid if KEGG groups share a high number of genes. Under this scenario, a significant motif density in two different groups can simply be the result of the presence of a high number of shared genes, and not of the conservation of RNA structure in the absence of sequence similarity. To avoid this situation, we first clustered the initial KEGG dataset into 23 disjoint (non-overlapped) sets (see Fig. S4 and Methods). From this set, we selected 19 non-overlapping groups that have significantly enriched RNA elements. Figure 5A shows a box plot chart containing the 3′-UTR motifs in the x-axis and the percentage of motif-containing mRNAs from each KEGG dataset in the y-axis. It is noteworthy that 15 out of 19 elements (78.9% of total cases) were statistically more abundant in mRNAs of the KEGG dataset from which the motif was initially identified. For example, the m00330 has a motif representation of ≈ 65% in the KEGG tcr00330 dataset, but less than 40% in all the remaining groups. Arrows above each motif represent the 15 significant motifs that are specifically enriched in their classes (group-specific motifs). Only in four cases the motif ranked at the second position: m00280, m0970, m03050 and m03010 (see below).

Figure 5 All-against-all comparison.

(A) Box plot representation of RNA motif density among the 19 non-overlapped datasets. Open circles indicate outliers groups where the identified motif was more abundant in the corresponding dataset compared with the remaining groups (a red circle indicates motif coverage in the original KEGG category). Black bar, second quartile (Q2, median); Box, interquartile range (IQR, Q3-Q1); Whisker, +/ − 1.5 IQR. (B) Heatmap of the previous chart showing three main clusters of RNA elements (I, II and III). A white/yellow color indicates high correlation, red color indicates low correlation.

Additionally, data were also visualized by a heatmap analysis using R package (Fig. 5B). The heatmap plot shows that [1] as anticipated, most of 3′ motifs were specifically enriched in the group from which they were derived (a white/yellow color indicates high correlation, red color indicates low correlation); [2] some elements displayed lower abundances: m00480 and m04120; and [3] other elements are widely enriched and have a high abundance in 3′-UTR mRNAs of metabolically unconnected groups: m00030, m00280, m04130 and m03010, being this last motif the one having the most widespread distribution. This evidence indicates that structured 3′-UTR elements can constitute a signal network, being mostly present in groups of mRNAs that belong to a given metabolic pathway (group-specific elements) or being part of a wide-ranging distribution. The dendogram depicted in Fig. 5B separates motifs into three main clusters. Cluster I contains 11 motifs involved in specific cellular processes. In addition, four elements from Cluster II are specifically enriched in the group from which they were discovered. However, it is noteworthy that these RNA elements (m00030, m00620, mm03020 and m04130) also present an important enrichment in the 3′-UTR of other groups. In this context, we speculate that certain cis-acting signals may be shared among different but related metabolic categories, i.e. m00030 (Penthose phosphate pathway) and m00620 (Pyruvate metabolism) thus organizing a combinatorial network of RNA-RBP interactions. Finally, Cluster III contains the global element m03010. The Top 2 RNA motifs extensively distributed, m00280 and m03010, also presented the highest abundances in our validation strategy using randomized datasets. This produces a high representation of these motifs among unconnected metabolic groups, which hampers the analysis of their biological relevance. Altogether, the estimation of motif density among groups that did not share any transcripts (19 non-overlapped datasets) showed that at least 15 group-specific elements were more abundant in their corresponding category compared with all remaining groups. Nevertheless, a few elements such as m00030 and m04130 were also enriched in other datasets. A similar analysis was performed using 27 metabolic groups sharing less than 25% of the genes and comparable results were obtained [85% of the RNA elements were group-specific (data not shown)].

Validating predicted motifs using independent genomic and gene ontology data sets

In previous sections we found that a number of RNA motifs are specifically enriched in groups of functionally interrelated coding genes. To obtain an independent validation of the representation of these motifs in functional categories, we performed a reverse validation in which we searched the complete T. cruzi genome with each of these 53 motifs, using the cmsearch algorithm. We next analyzed the positive hits for each motif to see if there was a significant enrichment in annotated functional categories in each group. To do this we fetched the results of these searches into the DAVID server (NIAID, National Institutes of Health) (Huang et al., 2007) to evaluate functional enrichment in each of the 53 lists of genome motif-containing targets. The results show that 73.6% of the putative elements (39 out of 53) were successfully used to predict related gene targets with the same biological function of the category from which the motif was originated (results available in Table S6).

To cross-validate the motifs, we used a dataset grouped according to the Gene Ontology project (http://www.geneontology.org/). We analyzed 93 groups of 3′ downstream sequences having at least 10 sequences each (see File S4). We found that 35 out of 38 motifs (92%) were over-represented in at least one GO category compared with random searches. As expected, 11 out of 35 elements (31%) were enriched in groups of closely related function: m00270, m00310, m00480, m00562, m03010, m03050, m03420, m03430, m03440, m04120 and m04130 (Table S7). These results show that ≈ 74% of the motifs analyzed here were successfully validated against a full search in the T. cruzi genome and a third part of the elements could also be confirmed using a different dataset grouped by ontology characters.

Co-occurrence of specific RBP recognition sites and regulon’s signals

When we inspected the list of mRNAs co-immunoprecipitated in TcDHH1-containing granules (Holetz et al., 2010) (cellular structures that appear to be more related to a RNA degradative rather than stabilizing process), we could not detect over-representation of any elements, with the exception of m00250 (Ala, Asp and Glu metabolism) (data not shown). This observation suggests that aggregation of transcripts in TcDDH1 granules may not be guided by specific cis-acting signals but for a general recruitment’s mechanism, still unknown.

We next searched for co-occurrence of experimental RNA-binding elements (previously described in our laboratory) and structured motifs identified in this study. To this end, we examined the targets of UBP1 and RBP3 -containing the recognition motifs UBP1m and RBP3m (Noe, De Gaudenzi & Frasch, 2008)- and counted the number of targets containing (or not) any of the candidate elements. The data obtained revealed that the RBP3 binding motif was enriched in the Ribosome KEGG group (tcr03010 dataset) (P < 0.05, χ2 test) (see Table 3). When UBP1 mRNA target hits were analyzed, we found that the UBP1 binding motif was enriched in six KEGG groups (m00230, m03420, m00020, m00010, m00785, m03450) compared with the entire T. cruzi genome representation (χ2 test, Bonferroni correction). Therefore, for seven categories a significant number of genes have a co-occurrence of both sequences (one specific RBP binding site and one predicted KEGG element) in their 3′-UTRs, suggesting that these particular groups could be coordinately regulated by specific trans-acting factors and thus defining a post-transcriptional regulon.

Table 3 Co-occurrence of specific RBP sites and regulon’s elements.

Co-occurence of structural KEGG RNA motifs and experimental RNA-binding sites of T. cruzi RBPs.

Motif	KEGG Description	N	%	Co-occ	P-value	B-H	Bonf.	
UBP1								
tcr00230	Purine metabolism	31	9.7	3	8.3E−09	8.3E−09	4.4E−07	
tcr03420	Nucleotide excision repair	21	9.5	2	3.0E−06	3.0E−06	1.6E−04	
tcr00020	Citrate cycle (TCA cycle)	24	8.3	2	1.6E−05	1.6E−05	8.3E−04	
tcr00010	Glycolysis/Gluconeogenesis	28	7.1	2	8.4E−05	8.9E−05	4.4E−03	
tcr00785	Lipoic acid metabolism	10	10.0	1	6.5E−04	7.1E−04	3.5E−02	
tcr03450	Non-homologous end-joining	10	10.0	1	6.5E−04	7.2E−04	3.5E−02	
tcr04650	Natural killer cell mediated cytotoxicity	11	9.1	1	1.3E−03	1.4E−03	6.7E−02	
tcr00650	Butanoate metabolism	13	7.7	1	3.5E−03	4.1E−03	1.9E−01	
tcr00630	Glyoxylate and dicarboxylate metabolism	14	7.1	1	5.3E−03	6.2E−03	2.8E−01	
tcr03018	RNA degradation	14	7.1	1	5.3E−03	6.4E−03	2.8E−01	
tcr03410	Base excision repair	14	7.1	1	5.3E−03	6.5E−03	2.8E−01	
tcr03440	Homologous recombination	15	6.7	1	7.5E−03	9.5E−03	4.0E−01	
tcr03430	Mismatch repair	18	5.6	1	1.7E−02	2.2E−02	9.2E−01	
tcr00310	Lysine degradation	21	4.8	1	3.2E−02	4.2E−02	1.7E + 00	
tcr00030	Pentose phosphate pathway	23	4.3	1	4.4E−02	6.0E−02	2.3E + 00	
RBP3								
tcr03010	Ribosome	186	3.2	6	3.0E−02	—	—	

RNA motifs are differentially expressed during parasite development and stress response

We used T. cruzi microarray data provided by the Tarleton laboratory (Minning et al., 2009) to investigate the motif representation among clusters of developmentally co-expressed genes. Firstly, we used the coXpress v1.3 program (http://coxpress.sf.net) to obtain 74 clusters of co-expressed genes. For each sequence in a cluster, we obtained 350 nts downstream of the stop codon of the CDS as annotated in TriTrypDB to obtain sequences resembling 3′-UTR (see Methods). Secondly, predicted 3′-UTRs of transcripts included in these groups were utilized to analyze the motif density for each putative RNA element. To analyze the statistical significance of the enrichment of these elements, we compared the experimental data against random distributions (P < 0.001, Z-test). Interestingly, 18 out of the 53 RNA motifs were statistically enriched in 20 groups of developmentally regulated (co-expressed) genes (see Fig. 6 and File S5). From these 18 elements, 13 (72%) belong to the 38 statistically significant candidates listed in Table 2: m00310, m00380, m00410, m00480, m00500, m00562, m00670, m00900, m00970, m03430, m03440, m04070 and m04120 (see Table 4 for a detailed list of developmentally regulated motifs).

Figure 6 Developmentally regulated cis-elements.

Profile expression of several developmentally regulated clusters having over-represented RNA elements (depicted at the right of the panels). A, amastigote, T, trypomastigote, E, epimastigote, M, metacyclic.

Table 4 Developmentally regulated motifs.

List of RNA motifs over-represented in clusters of developmentally regulated genes in T. cruzi.

Cluster	N	Functional enrichment	RNA motifs	
C1	166	nucleotide binding	m00100, m00130, m00310*,
m00350, m00563, m00670*, m04120*	
C3	96	WD40 repeat	m00500*	
C4	28	nucleoside binding	m03430*	
C5	56	Trypanosome sialidase	m00051, m00562*, m03440*	
C6	256	translation	m00480*	
C10	194	ribonucleotide binding	m00562*, m00563, m00670*	
C15	21	Trypanosome sialidase	m04070*	
C16	175	electron carrier transport	m00051, m00380*, m00410*, m00500*, m00563	
C17	96	Chaperonin Cpn60/TCP-1	m00130, m03440*	
C22	194	flagellar motility	m00563, m00670*	
C23	155	proteolysis	m00051, m00310*, m00670*, m00970*	
C24	22	—	m00900*, m04070*	
C26	16	sugar/inositol transporter	m00670*	
C28	34	—	m00562*	
C31	57	metal-binding	m00480*	
C32	28	Trypanosome sialidase	m00562*	
C40	11	—	m03440*	
C44	241	RNA polymerase act.	m00563	
C49	9	Trypanosome sialidase	m00562*	
C51	9	—	m00970*	
Notes.

N, number of sequences in each cluster.

* Statistically significant RNA motifs from Table 2.

Using a similar strategy we next investigated the transcriptional response of T. cruzi epimastigotes submitted to hyperosmotic stress. Using the data from Li et al. (2011), we analyzed the expression profiles of co-expressed genes and identified 33 sets of similarly regulated genes (File S6). As before, we analyzed the statistical significance of these element’s enrichments comparing the experimental data against random distributions (P < 0.001, Z-test). When epimastigote cells were subjected to hyperosmotic stress during a time-course experiment, nine RNA motifs were statistically over-represented in 10 co-regulated gene clusters (CI, CII, CIII, CVIII, CIX, CX, CXI, CXII, CXXII and CXXIII) compared to random searches (P < 0.001, Z-test). From these nine elements, six (m00310, m00510, m00562, m00670, m00785 and m03440) belong to the 38 statistically significant motifs (Table 5).

Table 5 RNA motifs over-represented in transcripts affected by stress.

List of structured RNA motifs over-represented in gene clusters affected under hyperosmotic stress conditions. CXXII contains non-regulated genes; CI, CII, CIII, CVIII, CIX, CXI contain up-regulated genes; and CXII, CXXIII contain down-regulated genes.

Cluster	N	Over-represented RNA motifs	
CI	95	m00480	
CII	64	m00310*, m00562*, m00670*	
CIII	61	m00562*	
CVIII	52	m00450, m00562*, m00785*	
CIX	42	m00510*, m00562*	
CX	34	m003440*	
CXI	60	m00310*	
CXII	63	m00510*, m03440*	
CXXII	53	m04144	
CXXIII	54	m03440*	
Notes.

N, number of sequences in each cluster.

* Statistically significant RNA motifs from Table 2.

There are four RNA motifs that are over-represented in four clusters of genes differentially up-regulated by stress. These elements are: m00310 (in CII and CXI), m00562 (in CII, CIII and CVIII), m00670 (in CII) and m00785 (in CVIII). On the other hand, two structured motifs were enriched in two clusters of down-regulated genes: m00510 (in CXII) and m03440 (in CXII and CXXIII). These results suggest that a number of candidate RNA elements identified in this work may be involved in the post-transcriptional regulation of a variety of genes whose expression changes significantly during the parasite’s life-cycle or upon stress.

Discussion

The life-cycle of an RNA in trypanosomatids mostly depends on post-transcriptional mechanisms due to the absence of a tight control at the transcriptional level. Given the large diversity of cellular transcripts, the separation of a transcriptome into modules of co-regulated genes is likely to be an advantageous strategy. Particularly in the case of trypanosomes, RNA regulons may represent an ideal system to achieve an adequate control of gene expression. In the past few years several reports provided evidence indicating that trypanosomal transcripts are organized as post-transcriptional regulons (Archer et al., 2009; Das et al., 2012; Estevez, 2008; Guerra-Slompo et al., 2012; Mayho et al., 2006; Noe, De Gaudenzi & Frasch, 2008). Furthermore, several RBPs have also been shown to interact with a subset of stage-specific mRNAs, suggesting the presence of developmental regulons (Dallagiovanna et al., 2008; Li et al., 2012; Mörking et al., 2012; Walrad et al., 2012).

Nowadays, the three-dimensional structure prediction tools of RNA based on its sequence constitutes a major challenge (Cruz et al., 2012). However, several methods for determining predictive models of RNA secondary structures have been described in the last decade (Seetin & Mathews, 2012) and, some of them, were used here to elucidate RNA regulons in ≈ 70% of the trypanosomal 3′-UTRs analyzed. Probably, the lack of advanced methods to facilitate the estimation of higher-order RNA structures is a partial explanation for why we failed to identify RNA motifs in ≈ 30% of the KEGG groups. Identification of conserved elements constitutes the primary step in the characterization of RNA regulons and to our knowledge, this is the first systematic genome-wide in silico screen to search for novel structural cis-acting elements in T. cruzi. The major findings of this investigation can be summarized as follow: [1] 38 KEGG groups have conserved structured elements mostly located in the 3′-UTR (Fig. 3); [2] these motifs have a preferred sense orientation and are positioned in the vicinity of the translational stop codon (Fig. 4); [3] structured RNA motifs found in the 3′-UTR are highly represented in its corresponding metabolic categories compared with random/remaining KEGG groups (Fig. 5), as previously reported in other RNA regulon heatmaps (Hogan et al., 2008). Through the binding of cognate RBPs, the RNA motifs present in a given metabolic pathway may guide the construction of regulatory networks, also called RNA regulons (see model in Fig. 7).

Figure 7 Model of conserved structural elements in T. cruzi.

Scheme showing the spatial organization of RNA motifs in 3′-UTRs of several transcripts. Only a single element is shown in each mRNA to simplify the figure, but a combinatorial organization (co-occurrence of different motifs) is required to ultimately describe the dynamics and connectivity of the post-transcriptional network. Numbers 1–15 represent genes belonging to different metabolic groups. RBP1-3, RBPs that specifically recognize RNA motifs (from group 1 to 3) and orchestrate three post-transcriptional regulons. In some circumstances, these interactions can occur in a developmentally regulated manner.

The element m03010 is greatly over-represented in the dataset, being found in ≈ 90% of the 3′-UTRs of the ribosomal protein genes. It was previously reported that ribosome biogenesis is controlled by post-transcriptional mechanisms (Grigull et al., 2004; Thorrez et al., 2008) and that sequence elements are shared by the transcripts encoding its components. For instance, a conserved sequence motif UUGUU is present in many ribosomal protein 3′-UTRs in nematodes (Hajarnavis & Durbin, 2006), probably involved in translation regulation. The data obtained revealed that the motif UUGUU is over-represented in the mRNAs of trypanosomal ribosomal proteins (92.7%, 280 of 302), compared to the remaining KEGG groups (75.5%, 780 of 1033) (P < 0.0001, χ2 test). This short motif is also present in the consensus sequence of the structural m03010 element.

A number of studies in trypanosomes show that genes encoding interrelated proteins have similar mRNA levels (Minning et al., 2009). For diverse groups of genes, this similarity in mRNA levels can be extended to concerted changes during differentiation or in perturbation experiments, suggesting that the transcriptome of these parasites is organized in clusters of transcripts exhibiting similar transcript abundance profiles (Ouellette & Papadopoulou, 2009; Rochette et al., 2008; Veitch et al., 2010). Also the abundance of certain ncRNAs varies between distinct forms of T. brucei (Michaeli et al., 2012). Here, we reported that functional groups in T. cruzi share common motifs (as depicted in Fig. 6), offering a starting point to screen for trans-acting factors in each set of mRNAs, or regulons, that probably modulate their abundance, turnover and/or access to the translation machinery.

Our results demonstrate that members of different clusters display similar RNA abundances in distinct stages of the T. cruzi life-cycle (Fig. 6 and Table 4) and in specific cellular conditions (Table 5). Analysis of developmentally regulated clusters with the DAVID Functional Annotation Chart tool revealed that clusters C1, C5, C23, C32 and C49 contain metabolically connected transcripts with profile expressions that might be coordinated by the over-represented elements (Table 4). For instance, the m00562 motif, present in the majority of genes of the inositol phosphate pathway follows in part the same profile expression pattern of Trypomastigote-like Trans-sialidase genes [notice that Trans-sialidase enzyme is GPI-anchored in the infective trypomastigote forms of the parasite (Chaves, Briones & Schenkman, 1993)]. This outcome could probably be the result of the same pattern of regulation exerted over a cohort of transcripts. These observations, together with the existence of common sequence signals involved in protein recognition within each group of mRNAs, make it possible to describe the components that make up numerous potential post-transcriptional regulons.

mRNP complexes are highly dynamic structures that can be rapidly remodeled in response to alterations in environmental conditions or during differentiation (Keene, 2007; Mansfield & Keene, 2009). Protein composition of these mRNP complexes modulates RNA biology in different environmental conditions or developmental stages. To assess whether post-transcriptional regulons are remodeled under stress conditions, we evaluated the abundance of structured RNA motifs in genes clustered by their expression profile after hyperosmotic stress during a time course study. Several clusters of stress regulated transcripts were observed (as depicted in Table 5), denoting that potential RNA regulons can still coordinate gene expression under osmotic stress conditions. In this context, it was also shown that TbDRBD3 remains bound to its specific target transcripts after starvation or arsenite treatment (Fernandez-Moya et al., 2012).

The four RNA elements that fulfill all the criteria examined in this work are m00310 (Lysine degradation), m00562 (Inositol phosphate metabolism), m00670 (One carbon pool by folate) and m03440 (Homologous recombination). These potential RNA-binding sites have a specific motif enrichment, a particular sense orientation, and are differentially expressed during parasite development and stress response. Although future experimental approaches are necessary to explore their biological functions, it is worth noticing that a complete in silico analysis was able to reveal hidden regulatory connections between these genes.

Regulatory cis-elements tend to be conserved among closely related organisms. To evaluate whether orthologous structural RNA elements could be acting in 3′-UTRs of other kinetoplastid parasites, we searched a variety of T. cruzi elements in T. brucei and L. major 3′-UTR datasets (unpublished data). As expected, several 3′ structured motifs identified in T. cruzi were also recognized in T. brucei [a species that diverged from the American parasite circa 100 million years ago (Stevens et al., 1999)], and to a lesser extent in L. major [speciation of Trypanosoma and Leishmania genus occurred 200–500 million years before present (Overath et al., 2001)]. These preliminary results suggest that there may also be a conserved phylogenetic signal in these structured RNA elements.

Conclusions

The regulon model states that RBPs coordinately regulate multiple mRNAs coding for interrelated proteins by interacting with transcripts containing shared elements. These post-transcriptional regulons could describe how gene expression is coordinately achieved in organisms where transcriptional regulation (at the initiation level) does not seem to play a major role. In this work, we reported the bioinformatic characterization of conserved structural cis-regulatory RNA elements in the 3′-UTRs of metabolically clustered T. cruzi transcripts. Using a computational approach, we have previously identified a collection of hundreds of RBPs encoded in the TriTryp genomes potentially involved in post-transcriptional mechanisms (De Gaudenzi, Frasch & Clayton, 2005). That genome-wide screen for RRM-type RBPs, is now followed in this work by the identification of novel putative RNA-binding sites shared by diverse mRNAs. Taken together, these two computational studies lay out the foundation required for further functional characterization of these post-transcriptional regulons in trypanosomatids.

Methods

Databases

The T. cruzi database (T. cruzi CL Brener genomic sequence Release 5.1) utilized in this work was obtained from TriTrypDB Database Resource (www.tritrypdb.org) (Agüero et al., 2006). 5′ upstream genomic sequences and 3′ downstream genomic sequences were obtained using the TriTrypDB sequence retrieval tool. A length of 80 nts upstream to the CDS was used to obtain sequences resembling the 5′-UTR, while 350 nts downstream to CDS were used for 3′-UTR, in agreement to previously reported data from trypanosomes (Campos et al., 2008). The 5′-end of transcripts was estimated as a 350-nt sequence encoding the predicted 5′-UTR followed by the first 270 nts of the coding region (in order to cover the same total length of the 3′-UTRs).

Trypanosomatid genes were grouped by the KEGG pathway database (Kanehisa & Goto, 2000). EST databases (T. cruzi filtered) were downloaded from NCBI and BLASTn searches were performed with the following parameters “−F F −W 7 −E 1e−5 −S 1 −b 1 −v 1 −m 8”. DAVID Functional Annotation Chart tool (Huang et al., 2007) (http://david.abcc.ncifcrf.gov/) was used to categorize and compare the different gene lists against a T. cruzi background (using Fisher’s exact test and Benjamini-Hochberg correction).

RNA motif elucidation

All computational analyses were performed using free software, available in the public domain and compiled for a LINUX environment (Ubuntu 9.10 distribution). Using CMfinder stand-alone software version 0.2 (Yao, Weinberg & Ruzzo, 2006) (http://bio.cs.washington.edu/yzizhen/CMfinder/), common elements were identified in 53 different datasets containing at least 10 sequences, using the following parameters “−s1 −f 0.6 −c10”. As a control, motifs were also obtained using random sets. The 53 candidate motifs obtained were used to build stochastic context-free grammar (SCFG) model with the Infernal program (Nawrocki, Kolbe & Eddy, 2009). These models are representations of the RNA secondary structure and were used as queries in more refined searches (cmsearch algorithm). To validate the usage of models, a full test comparison between cmsearch and CMfinder’s output was performed to determine the parameters of true and false positive rates. Thus, each model was used to assess the element coverage within the 3′ sequences in the KEGG dataset (m00010 vs. tcr00010, m00020 vs. tcr00020, etc). We found that cmsearch tool gave a very good performance, being able to find most of the originally predicted CMfinder elements (mean “sensitivity” 0.91), giving very few false positives (mean “1-specificity” of 0.13) with a mean accuracy of 0.90.

Randomization test

Random datasets were constructed using a custom Perl script (version 5.8). Fifty different groups containing 3′-UTRs were obtained by randomly shuffling the original dataset of 1814 sequences from the KEGG repertoire and used in searches with CMfinder and Infernal programs. Each KEGG motif was used to search with Infernal program (Nawrocki, Kolbe & Eddy, 2009) against 50 random groups containing a similar number of sequences.

Statistical analysis

All statistical tests were corrected for multiple-testing, using the Benjamini-Hochberg False Discovery Rate (FDR) procedure. To test statistical significance of detected motifs, we first evaluated the specific enrichment of a given element. We defined specific enrichment as the mean difference of sequences having the motif in the original set, compared to sets of non-related sequences. For this, we generated a distribution of specific-motif enrichment by searching a given motif into 50 groups of non-related 3′-UTRs (randomly generated groups). The motif coverage (i.e., the number of sequences having a motif divided the total number of sequences in the group) was normally distributed, allowing us to calculate a Z-score for each putative motif [(specific-motif coverage — mean coverage in random groups)/standard deviation]. As the expected motif coverage was sensitive to the size of the group, we actually generated distributions for different group sizes (N = 5 to N = 200) and evaluated each motif significance with the corresponding distribution (according to its size). To determine if the specific-enrichments are significant, we generated a null distribution of motif-specificity, from sets of non-related sets of UTRs. A second Z-score was calculated and used to determine whether a particular specific-enrichment is expected by chance or not. Because this test was more restrictive, and to avoid losing too much sensitivity, we set an FDR threshold of 0.1 (i.e., 10% of false positives are expected). Motif candidates were further filtered based on a motif coverage test. The motif coverage was calculated for each KEGG group using Infernal algorithm [using parameter “−toponly −E 1” (Nawrocki, Kolbe & Eddy, 2009)]. Then, we tested if the motif coverage in a given set of UTRs is significantly higher compared to the set of complementary strands (i.e., non-transcribed strands where no motifs are expected). For motif detection in non-transcribed regions, the parameter “−bottomonly” was applied. Statistical significance of the coverage difference between UTRs and non-transcribed sequences (sense vs. anti-sense) was assessed by a χ2 test (FDR 5%).

Sequence logos and secondary structures

The motif logo was constructed using WebLogo (http://weblogo.berkeley.edu). RNAfold tool (http://rna.tbi.univie.ac.at/) (Hofacker, 2003) was used to plot the secondary structure of the 53 predicted RNA motifs.

Gene filtering

Many T. cruzi genes are present in many copies. To avoid such sequence bias in motif detection procedures, we filtered out paralogous genes whose UTRs have >80% of sequence identity (BLAST alignment with at least 280 of the 350 nucleotide matches). In addition, truncated UTRs (<350 nts) and UTRs having a stretch of 5 or more ambiguous nucleotides (NNNNN) were discarded. A total of 106 genes were eliminated after applying these filters using custom Perl scripts.

Generation of disjoint (non-overlapping) KEGG groups

In order to unbiasedly test if some motifs are present in more than one KEGG group, we first defined non-overlapped KEGG groups (i.e., groups that have no genes in common). A measure of overlap between a pair of KEGG groups was calculated as the number of shared genes divided by the number of total distinct genes considering both groups (set intersection/set union). Figure S4 shows a dendogram of KEGG groups after a complete-linkage clustering. All branches starting at a height of 1 are disjoint (non-overlapped), while groups joining at lower heights have increasing degrees of overlap. To make a dataset of non-overlapped groups, we simply selected one KEGG group of each branch.

Clustering

Hierarchical clustering was carried out using CoXpress 1.3 software (http://coxpress.sf.net). Briefly, 5268 genes obtained from T. cruzi microarray data previously published by the Tarleton lab (Minning et al., 2009) were clustered into 74 profiles using the following parameters: h = 0.05, r = 0.95. Stress data was extracted from Table S2 of Li et al. (2011) collected by the Docampo lab. Concisely, 1468 genes affected after hyperosmotic stress were clustered into 33 profiles using the following parameters: h = 0.01, r = 0.95. RNA motifs were used to search against clustered and 50 random groups using cmsearch algorithm and statistical significance was calculated by Z-test. Abbreviation List

ARE AU-rich element

mRNP messenger ribonucleoprotein

RBP RNA-binding protein

RRM RNA-recognition motif

UTR untranslated region

Supplemental Information

Figure S1 Relative frequency of 3′ motifs in 5′-end and 3′-UTR KEGG datasets

Frequency of motifs (total hits/kb) in KEGG groups fractioned in two subsets composed of 5′-end and 3′-UTR. Searches were performed using the elements identified in the 3′-UTR as queries against the estimated 5′-end of transcripts (see Methods). *, statistically significant RNA motifs from Table 2.

Click here for additional data file.

Figure S2 Motif orientation

Percentage of motifs located in sense (blue) and anti-sense (red) orientation in the 3′-UTR. RNA elements marked with a black circle have no statistical significance (χ2 test, FDR 5%).

Click here for additional data file.

Figure S3 Number of genes of each group having RNA motifs within annotated transcripts

Chart of the percentage of sequences harboring the motif within the EST hit in each of the 53 KEGG groups analyzed here. Overall, 43 out of 53 categories have at least 50% of the BLAST sequences containing the complete motif within the EST hit, reinforcing the idea that our datasets could be used to identify putative regulatory RNA elements.

Click here for additional data file.

Figure S4 Non-overlapped dataset definition

The input metabolic dataset, 53 KEGG groups, were hierarchically clustered into 23 different branches that do not share any gene with any other branch. The schematic representation shows a graphic with a height scale that varies between 0 (completely overlapped sets) to 1 (totally different sets). Next, we selected 19 groups with non-overlapped genes which contain significant conserved RNA motifs (marked with asterisks) to perform a comprehensive analysis of RNA motif representation (see Fig. 5).

Click here for additional data file.

File S1 KEGG dataset

List of 3′-downstream genomic sequences used for motif elucidation. Fasta file of 3′-downstream sequences of the 1617 genes from the 53 T. cruzi metabolic KEGG groups.

Click here for additional data file.

File S2 Sequence LOGO and secondary structure notation of the 53 RNA motifs

Sequence alignments of the 53 cis-elements obtained for each set of sequences were used to plot motif logos using WebLogo version 2.8.2 (2005-09-08) (http://weblogo.berkeley.edu/) (Crooks et al. 2004). Five elements contain U-rich sequences (m00260, m00500, m00900, m03010 and m04144), 14 have AU-rich sequences (m00010, m00020, m00190, m00230, m00250, m00270, m00520, m03018, m03030, m03040, m03050, m03410, m03420 and m04650) and three are enriched in GU nts (m00051, m00970 and m03020). Nonetheless, most of the structured elements did not show a preferential enrichment at the primary sequence level and have a variable nucleotide composition.

Click here for additional data file.

File S3 List of mRNAs bearing motifs

Full list of 1335 sequences having the different structured elements. Number, No. of gene; GeneID, Systematic gene name; motif, motif’s name; motif˙star, the numbers indicate distances from translation stop codon where the motif stars; motif˙end, position where the motif ends; motif˙length, length of the motif in nucleotides; gc˙content, percentage of GC within the motif.

Click here for additional data file.

File S4 Gene ontology clusters.

List of 93 GO groups’ genes.

Click here for additional data file.

File S5 Developmentally regulated clusters

GeneID list of developmentally regulated clusters depicted in Fig. 6.

Click here for additional data file.

File S6 Clusters of genes involved in stress response

GeneID list of 33 gene clusters affected by hyperosmotic stress.

Click here for additional data file.

Table S1 Motif representation in predicted 3′-UTRs of 53 T. cruzi metabolic groups

The 3′-UTRs from the 53 T. cruzi KEGG groups were used to search statistically significant RNA motifs with CMfinder program. Column 2, motif coverage (%) in the different 3′- UTR datasets; Column 3, motif coverage in 50 random datasets (mean + /− sd); Column 4, P-values of the specific-motif enrichment test; Column 5, P-value after BH FDR 10%. In red, 11 not statistically significant RNA elements. In black, 42 statistically significant RNA motifs.

Click here for additional data file.

Table S2 Motif orientation in the 3′-UTR

Percentage of motifs located in sense (column 2) and anti-sense (column 3) orientation in the 3′- UTRs. N, total number of sequences; Nro. SE, number of sense motif-containing sequences; Nro. AS, number of anti-sense motif-containing sequences. P-value, chi-square test. In red, not statistically significant RNA elements.

Click here for additional data file.

Table S3 Motif position in the ESTs

KEGG genes having the motif located in the EST-seq (in blue) or not (in red). The key is within the file.

Click here for additional data file.

Table S4 Motif localization within 3′-UTR

Percentage of motifs located in the first fragment of the 3′- UTR (less than 175 nts of distance from the translation stop codon).

Click here for additional data file.

Table S5 All-against-all comparison

Motif density of all RNA elements in 3′-UTRs of the 53 KEGG categories.

Click here for additional data file.

Table S6 Genome targets

Table of motif-containing genes in the T. cruzi genome. Motif name— Description— Category— Term— Count— %— PValue— Genes— List Total— Pop Hits— Pop Total— Fold Enrichm.— Bonferroni— Benjamini— FDR— In red, no-enriched targets genes.

Click here for additional data file.

Table S7 GO targets

Table of structured RNA motifs statistically enriched in a given GO dataset.

Click here for additional data file.

We are indebted to Dr. Adriana Jäger for helpful discussions and critical reading of the manuscript. We also thank the reviewers of this manuscript for their beneficial comments and suggestions.

Additional Information and Declarations

Competing Interests

Author Contributions

The authors declare there are no competing interests.

Javier G. De Gaudenzi conceived and designed the experiments, performed the experiments, analyzed the data, wrote the paper.

Santiago J. Carmona performed the experiments, analyzed the data, wrote the paper.

Fernán Agüero analyzed the data, wrote the paper.

Alberto C. Frasch conceived and designed the experiments, analyzed the data, wrote the paper.

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
