# Peer review of "Genome-wide analysis of 3′-untranslated regions supports the existence of post-transcriptional regulons controlling gene expression in trypanosomes"

_PeerJ, doi:10.7717/peerj.118_

## Round 0.1 · original submission · Minor Revisions

This paper should be acceptable for publication after some minor re-writing and typographic fixes.

Reviewer 1 ·

Basic reporting

No Comments

Experimental design

No Comments

Validity of the findings

No Comments

Additional comments

Following a rigorous controls and robust statistical analyses, the authors detect the existence of conserved structured RNA motifs for different subset of related genes in Trypanosoma cruzi. The results are quite convincing.

There are some minor issues to address:

1) Contents of suppl. figure 1 and 2 are switched

2) Names of the motifs in suppl figure 2 include a “cr” that is not included in the rest of the manuscript

3) In lines 148-7 it is stated that: “To see if 3′ motifs are specifically located at 3′-UTRs, the searches were repeated but employing T. cruzi 5′-UTR and CDS grouped according to KEGG pathways.”, indicate how the length differences among 3′-UTR, 5′-UTR and CDS were considered for the motif abundance comparison purpose

4) In lines 149-151 it is stated that: “With the exception of m03010 the abundance of the in 5′-UTR and CDS datasets was lower than 15% and 10% respectively (see Supplemental Fig. S1)”, check the values since, as shown in the figure, there are some 3′ motifs whose abundance at the 5′-UTR is above 15% (i.e. m00563 and m04130) and at the CDS are also above 15% (i.e.m00100, m00310, m00410, m00450)

5) In the legend to suppl. file 3 there is a “t” missing: “the numbers indicate distances from translation stop codon where the motif stars”

6) In lines 197-9 there is a duplicated sentence: “However this examination is not valid if KEGG groups share a high number of genes.”

7) It would be convenient to include the label for Y Axis in figure 5a

8) The reference in the text to suppl. file 5 should be moved from line 281: “Firstly, we used the coXpress v1.3 program (http://coxpress.sf.net) to obtain 74 clusters of co-expressed genes (Supplemental File S5).” To line 287: “Interestingly, 18 out of the 53 RNA motifs were statistically enriched in 20 groups of developmentally regulated (co-expressed) genes (see Fig. 6).” since it contains the data for the 20 groups of developmentally regulated (co-expressed) genes

9) In table 4 indicate the meaning of N and explain the two N values within some groups (i.e. C6, C44)

10) Indicate the meaning of N also in table 5

11) In table 5 there are extra “0” for the motifs: m03440 and m04144

12) In addition, while in the text it is establish that (lines 297-300): “When epimastigote cells were subjected to hyperosmotic stress during a time-course experiment, seven RNA motifs (m00310, m00450, m00510, m00562, m00670, m00785 and m03440) were statistically over-represented in 10 co-regulated gene clusters (CI, CII, CIII, CVIII, CIX, CX, CXI, CXII, CXXII and CXXIII) compared to random searches (P<0.001, Z-test) (Table 5).” the motifs m00480 and m04144 are also included in the table as over-represented RNA motifs

13) In line 301, it should be desirable to indicate the motif that does not belong to the 38 statistically significant motifs and/or modify table 5 similarly to table 4 to indicate this

14) It would be helpful for the reader to explicit in the text, or in the legend of table 5, which of the 10 co-regulated clusters are down and which are up-regulated. Indeed it is not easy to interpret the text in line 302-4: “There are four RNA motifs that are over-represented in three clusters of genes differentially up-regulated by stress. These elements are: m00310 (in CII and CXI), m00562 (in CII, CIII and CVIII), m00670 (in CII) and m00785 (in CVIII).” Which of the clusters indicated are the three clusters of genes differentially up-regulated by stress?

15) In line 301 there is an extra “V”: “and m03440 (in CXVII and CXXIII)”

Reviewer 2 ·

Basic reporting

This paper is very well-written and clear with minor exceptions that are listed below. The figures are suitable and understandable to a general reader. I believe the manuscript was prepared in keeping with criteria and polices of PeerJ.

Experimental design

The design is largely computational, and I believe it is suitable and critically performed. More detail is in my comments for the authors.

Validity of the findings

I believe the criteria for meeting the standards were met by these authors, and that the validity of the data is sound.

Additional comments

This is a very interesting and timely study of predominantly structural 3’UTR cis elements that share common locations in mRNAs that encode functionally related proteins during growth and development of T. cruzi. The cis mRNA binding sites for trans-acting factors such as RNA-binding proteins and noncoding RNAs represent a platform for the combinatorial codes of RNA regulons. The authors have used available tools to devise an algorithm (flow chart) of discovery of coordinated mRNA subpopulations. Among the key aspects of this work is that computational tools to discern RNA structural elements were applied here and yielded new and exciting information that is directly applicable to known metabolic pathways in T. cruzi (and are conserved in other trypanosome species as well). Thus, the data take a very different approach to the elucidation of RNA regulons in this organism, many of which reach have reached very similar conclusions. However, this not only confirms other potential RNA regulons, but also reveals several new potential examples. In total, trypansomes are quickly emerging as the best studied and best understood biological system in which RNA regulons dominate the outcomes of gene expression. This paper greatly extends this evidence and our understanding of gene expression that is applicable to other species.

Suggestions:

1. I would like to see a more in-depth discussion (a few sentences) about how advanced methods for determining higher-order RNA structure are lacking but that reliable one exist for secondary structural analysis. In spite of this limitation, the authors are able to apply these available methods to demonstrate common secondary structural elements among mRNA subsets. As methods for predicting tertiary and higher-order RNA structures improve they are likely to reveal many more examples of global RNA coordination though combinatorial cis-trans interactions. The lack of advanced methods of RNA structural analysis is also a partial explanation for why it may not be possible at this time to readily elucidate RNA regulons in more of the 53 KEGG groups. But the algorithm in figure 1a may be applicable for this in future years as well. In this sense, this paper is pioneering.
2. With respect to motif enrichment, the numbers are impressive. But I am surprised that there are very few (15%) cis-acting structural motifs in 5’UTRs. This is consistent with the fact in other species that most regulation of mRNA stability tends to involve 3’UTRs, but 5’ structures are often important for translational regulation. Alternatively, this may suggest that RNA regulons, in general, tend to involve 3’ UTRs. Do the authors believe this is characteristic of trypansomes more than other species?
3. There is a repeated sentence at line 198 on page 7.
4. Page 13 line 376: Use of the terms “motifs”, is unclear. It seems to me that “four motifs” should be changed to something like “four hallmarks” or “four criteria”, or? But it confuses the other meaning of the word motif in the paper.
5. Figure 4: this needs more explanation. Please give a clearer interpretation of the probability density function. What is the trend of these distributions and what is the basis for assigning the 175 nucleotide spot? Do the low frequency numbers at 300 and 340 in figure 4a indicate a declining trend toward a boundary? Is there a boundary at the stop codon?
6. Figure 5 is very convincing and similar to other RNA regulon heatmaps (e.g. Hogan, Brown et al., PLoS Biology 2008). However, Cluster II is a bit unclear. What do the authors mean by “more broad distributed elements” (P. 8 line 223)? Are these interpreted as not being RNA regulons?
7. Figures 6 and 7 are excellent and really drive the points home.

---

## Round 0.2 · accepted · Accept

Congratulation; I look forward to seeing the article published.